# Viral Equine Encephalitis, a Growing Threat to the Horse Population in Europe?

**DOI:** 10.3390/v12010023

**Published:** 2019-12-24

**Authors:** Sylvie Lecollinet, Stéphane Pronost, Muriel Coulpier, Cécile Beck, Gaelle Gonzalez, Agnès Leblond, Pierre Tritz

**Affiliations:** 1UMR (Unité Mixte de Recherche) 1161 Virologie, Anses (the French Agency for Food, Environmental and Occupational Health and Safety), INRAE (French National Institute of Agricultural, Food and Environmental Research), Ecole Nationale Vétérinaire d’Alfort, Université Paris-Est, 94700 Maisons-Alfort, France; muriel.coulpier@vet-alfort.fr (M.C.); cecile.beck@anses.fr (C.B.); gaelle.gonzalez@anses.fr (G.G.); 2RESPE (Réseau d’épidémio-surveillance en pathologie équine), 14280 Saint-Contest, France; Stephane.pronost@laboratoire-labeo.fr (S.P.); pitritz@wanadoo.fr (P.T.); 3LABÉO, 14280 Saint-Contest, France; 4BIOTARGEN, UNICAEN, NORMANDIE UNIV, 14000 Caen, France; 5UMR EPIA (Epidémiologie des Maladies Animales et Zoonotiques), INRAE, VetAgro Sup, Université de Lyon, 69280 Marcy L’Etoile, France; agnes.leblond@vetagro-sup.fr; 6Clinique Vétérinaire, 19 rue de Créhange, 57380 Faulquemont, France; 7AVEF (Association Vétérinaire Equine Française), Committee on Infectious Diseases, 75011 Paris, France

**Keywords:** encephalitis, arbovirus, rabies, Equid herpesviruses, Borna disease virus, West Nile virus, horses

## Abstract

Neurological disorders represent an important sanitary and economic threat for the equine industry worldwide. Among nervous diseases, viral encephalitis is of growing concern, due to the emergence of arboviruses and to the high contagiosity of herpesvirus-infected horses. The nature, severity and duration of the clinical signs could be different depending on the etiological agent and its virulence. However, definite diagnosis generally requires the implementation of combinations of direct and/or indirect screening assays in specialized laboratories. The equine practitioner, involved in a mission of prevention and surveillance, plays an important role in the clinical diagnosis of viral encephalitis. The general management of the horse is essentially supportive, focused on controlling pain and inflammation within the central nervous system, preventing injuries and providing supportive care. Despite its high medical relevance and economic impact in the equine industry, vaccines are not always available and there is no specific antiviral therapy. In this review, the major virological, clinical and epidemiological features of the main neuropathogenic viruses inducing encephalitis in equids in Europe, including rabies virus (*Rhabdoviridae*), Equid herpesviruses (*Herpesviridae*), Borna disease virus (*Bornaviridae*) and West Nile virus (*Flaviviridae*), as well as exotic viruses, will be presented.

## 1. Introduction

Neurological disorders represent an important sanitary and economic threat to the equine industry worldwide. Even mild nervous deficits can result in poor performances and long recovery of athletic horses, while severe clinical signs can induce life-threatening injuries in infected horses and may expose owners, veterinarians and care providers to significant risks [1]. Few surveys have been carried out to evaluate the burden of neurological diseases in horses and were performed almost 20 years ago. They indicated that neurological affections accounted as the fifth cause of death (8%) in adult horses, behind foaling (24%), digestive (21%), locomotor (21%), and cardiovascular (9%) causes. In two studies performed in Australia and in France, neurological diseases were first attributed to trauma (26% to 34%), congenital malformations (19% to 20%), while inflammation and infection were reported in 6% to 17% of horses with neurological conditions [2,3,4]. Early recognition of neurological infectious diseases may increase the chance of a positive outcome and is key to the implementation of coordinated management measures designed to prevent large-scale outbreaks when highly contagious pathogens, such as equid herpesviruses, are involved. Many neurotropic viruses affecting equines are also significant human pathogens and rapid identification of zoonotic viruses in horses is pivotal in their surveillance and in the control of corresponding human viral diseases [5].

Multiple neuropathogenic pathogens, either viruses, bacteria or protozoa, can induce an important inflammation of the central nervous system (CNS). Bacterial meningitis are common neurological infections in foals, while neuroborreliosis, listeriosis and Equine Protozoal Myeloencephalitis are rare and difficult to diagnose in equines [1]. As far as viruses are concerned, *Rabies virus*, *Equid alphaherpesvirus 1* (EHV-1), *West Nile virus* and related flaviviruses (*Japanese Encephalitis virus, Saint-Louis Encephalitis virus* and *Murray Valley Encephalitis virus*), *Mammalian 1 orthobornavirus* and neurotropic alphaviruses (*Eastern, Western and Venezuelan Equine Encephalitis Virus* species) are the most largely described neuropathogenic viruses (Figure 1).

Equine neuropathogenic viruses generally induce encephalitis or myeloencephalitis, which is an inflammation of the central nervous system (cortex, brain stem, and cerebellum) and/or of the spinal cord characterized by large or multifocal infiltrations of mononuclear cells (Figure 2d). Infected animals may experience behavioral change, as well as balance, posture and gait deficits (Figure 2a–c) [1]. Neurological examination, including testing of reflexes, evaluating postures and movements, is key in the clinical approach and allows assessing the course of disease and therefore its prognosis and response to therapeutic options. However, it is worth to note that such neurological examination and scoring is difficult to standardize—even among highly specialized practitioners [8]. In addition to posture and gait disorders, hyperthermia and sudden clinical signs peaking after 48 h of infection, guide viral encephalitis diagnosis. Usually, high fever is considered as a warming sign even if infectious diseases are not the only reason of hyperthermia and if they are not systematically detected during the horse clinical examination. Indeed, 14% to 38% of West Nile disease diagnosed in Europe and 52% of equid herpesvirus myeloencephalopathy (EHM) cases evidenced in France had hyperthermia during veterinary examination [9,10,11]. Cerebrospinal fluid (CSF) findings will generally be informative of a viral meningo-encephalitis, comprising an increased protein concentration, normal glucose concentration and pleomorphic leucocytosis with predominating mononuclear cells or neutrophils [12].

Epidemiological parameters including knowledge on the holding conditions, horse condition and nutritional needs, prevalent pathogens in a specific region (Figure 3) will help to prioritize the hypothesis. Updated epidemiological data are strongly needed, such as the ones provided by national surveillance systems such as RESPE (Réseau d’épidémio-surveillance en pathologie équine) in France [13] or EQUINELLA in Switzerland [14] ). In Europe, many viruses, including two zoonotic viruses, should be recognized promptly: rabies virus is one of the most important global zoonotic pathogen and has been eradicated from Western Europe by successful vaccination campaigns, while West Nile virus is a (re)emerging arthropod-borne virus that has recently spread in Europe to the Balkans area and northern-most countries (Germany) [15,16,17]. From an economic perspective, Equid herpesviruses (EHV-1 in particular) are one of the most important equine pathogens in Europe [18]. In France, EHV-1 was the principal cause of neurological infections from 2008 to 2011, with 26 cases of EHM over 214 neurological cases reported (12%) [10].

Most equine neuropathogenic viruses will induce similar clinical presentations and laboratory testing, either by indirect (ELISA, seroneutralisation or other serological assays) or direct assays (PCR, virus isolation, staining of viral antigens from infected tissues) must be provided in order to confirm the etiology of the disease (Table 1). For the past few years, direct methodologies, especially for EHV-1 detection, have been promoted. However, for most arthropod-borne viruses as well as for Borna disease virus, indirect assays have sustained interest owing to low and short-lasting viremia in infected horses. Serological screenings are also widely used in epidemiological surveys for the surveillance of equine encephalitis worldwide. Since horses can be vaccinated against most equine neuropathogenic viruses (with the notable exceptions of Borna disease and some exotic equine encephalitis viruses) (Table 2), the immunization status must be known for the interpretation of the serological tests. In this review, we will present the major virological, clinical and epidemiological features of the main neuropathogenic viruses inducing encephalitis in equids in Europe, namely rabies virus (*Rhabdoviridae*), equid herpesviruses (*Herpesviridae*), Borna disease virus (*Bornaviridae*) and West Nile virus (*Flaviviridae*), as well as exotic viruses. The most relevant information on the diagnosis and prevention from equine encephalitis viruses enzootic in Europe will also be presented in this review.

## 2. Equine Encephalitis Viruses Enzootic in Europe

Equine viruses causing encephalitis can be classified into two groups: viruses transmitted indirectly through the bites of an infected arthropod (mosquitoes, ticks, or midges) or by direct transmission (Figure 3). Direct transmission viruses are usually associated with highly contagious horse-horse contacts, secretions or excretions (with the prime example of EHV-1 infected horses) (Figure 3a) compared to arthropod-vector transmitted neuropathogenic viruses; most equines infected by neurotropic arboviruses are indeed considered as dead-end hosts, with the remarkable exception of the epizootic strains of Venezuelan equine encephalitis virus, an arbovirus only identified in America (Figure 3b) [20,21].

In this section, we will address several viral equine encephalitis inducing the most important clinical and economic consequences in Europe: rabies virus (*Rhabdoviridae*), Equid herpesviruses *(Herpesviridae*), Borna disease virus (*Bornaviridae*) and neurotropic flaviviruses transmitted by mosquitoes and ticks, e.g., West Nile, tick-borne encephalitis and Louping ill viruses (*Flaviviridae)*.

### 2.1. Equid Herpesviruses

Highly successful pathogens of horse populations worldwide, Equid herpesviruses induce latent infections that may cause abortions, respiratory (rhinopneumonia), and neurological diseases (myeloencephalopathy). Nine herpesviruses have been described in the family Equidae, which includes horses, ponies, donkeys and zebras and two natural hosts have been identified key for EHV epidemiology: horses for equid herpesviruses 1 (EHV-1), EHV-2, EHV-3, EHV-4, and EHV-5 and donkeys for EHV-6, EHV-7, and EHV-8 [22]. Concerning EHV-9, zebras, as well as the African rhinoceros, could serve as virus reservoirs [23]. Equid alphaherpesvirus 8 (EHV-8), formerly known as asinine herpesvirus 3, was recently considered as a new threat to the horse industry; it was shown to cause abortion in horses in Ireland and was associated with one neurological case in a donkey in China [24].

Equid alphaherpesvirus 1 (EHV-1) was recognized as a neuropathogenic virus in horses in 1966 [25]. Within the last 20 years, EHM has been considered as an uncommon sequela of EHV-1 infection in horses that led the USDA to classify EHM as a re-emerging disease. If different cases of EHM caused by EHV-4 were strongly suspected, no cases have been reported in the literature up to now. Although EHV-1 and EHV-4 share a high degree of genetic and antigenic similarities, differential virus tropism (the former virus being more endotheliotropic than EHV-4) and ability to interfere with the innate immune response could explain the differences in host range and pathogenicity between EHV-1 and EHV-4, as suggested by Ma et al. [26]. Nevertheless, similar to sporadic cases of EHV-4 abortion, it cannot be ruled out that neurological cases may occur with EHV-4.

Virus: EHV-1 and 4 are members of the *Varicellovirus* genus, in the *Alphaherpesvirinae* sub-family in the family *Herpesviridae* (Figure 1). These alphaherpesviruses are characterized by lytic infection and can establish a lifelong latent infection in blood circulating and lymph node-residing lymphocytes, as well as in sensory neurons within the trigeminal ganglia, which may reactivate upon stress [27,28]. The linear double-stranded DNA genome of EHV-1 contains 80 open reading frames and is 150 kb long and consists of a unique long (UL) and unique short region (US) [29].

Transmission and epidemiology: Virus transmission occurs through direct contact between horses, fomites, infectious aerosols and/or indirectly by humans. A recent study demonstrating the survival of EHV-1 in water strongly suggests a potential new way of transmission [30]. EHV-1 causes frequent outbreaks of abortion and myeloencephalopathy worldwide, even in vaccinated horses (Figure 4). EHV-1 outbreaks have been reported for centuries and many cases are reported across Europe, in France, Great Britain and Belgium, in the United States, in New-Zealand, Australia, Chile, Argentina, Israel and United Arab Emirates [31,32,33]. EHM incidence has increased in most parts of the world, in Europe and North America, as well as in Oceania, Africa, and Asia [10,11,28].

Pathogenesis and clinical disease: After primary replication in the respiratory tract, EHV-1 disseminates via cell-associated viremia in peripheral blood mononuclear cells and subsequently infects the endothelial cells of the pregnant uterus or central nervous system, leading in some cases to abortion and/or neurological disorders [34]. The incubation period of the disease is 6–8 days (before neurological signs become apparent), both in experimentally and in naturally EHV-1-infected horses [35]. Histologically, the most frequently observed lesion of EHM is vasculitis in the brain and/or spinal cord, leading to brain damages by hypoxia [36,37]. Neurological signs of EHM range from temporary ataxia, paresis, loss of sensation around the tail and perineal area and urinary incontinence to complete paralysis and death [38]. Affected horses may recover completely, while recumbency often leads to a fatal outcome [39]. The increased interest of researchers in the manifestations of this disease is not only due to the lack of current scientific understanding but also to the associated economic impact [40,41]. Infections that cause severe neurological dysfunction may only involve either one or two horses [42,43] or be associated with larger outbreaks [42,44]. Neurological syndromes have also been observed in various environments open to horses: breeding farms, riding schools, racetracks and, more recently—veterinary hospitals. Furthermore, while several breeds and age groups seem to be at a lower risk of developing EHM [42,45], other factors, including EHV-1 strains, host immunity and still unknown parameters, could explain why experimental EHV-1 infections were frequently partially successful [46,47].

Interestingly, Nuggent et al. and Allen et al. showed in 2006 that a single point mutation of adenine to guanine at nucleotide position 2254 in the catalytic subunit of the gene encoding DNA polymerase (ORF30) was often associated with EHV-1 neuropathogenicity (in 83% to 86% of cases), while absent in the majority of EHV-1 abortion outbreaks [48,49]. The non-synonymous A to G substitution at nucleotide position 2254 results in the replacement of asparagine (N) in position 752 (N752→D752). Experimental infections with recombinant viruses performed by Goodman et al. (2007) demonstrated that the N752 sequence variant of EHV-1 DNA Pol, when compared to the D752 variant, generated a low level of viremia in natural hosts and presented with reduced overall pathogenicity and capacity to induce neurological signs [46]. The discovery of this single polymorphism in ORF30 led to the development of a SNP-PCR (SNP—single nucleotide polymorphism) test for the detection of the two genotypes (potential neuropathogenic and non-neuropathogenic strains) [50]. Many studies performed in the field in different countries to characterize the neuropathogenic and non-neuropathogenic variants of Equid alphaherpesvirus 1, demonstrated neurological cases with A2254 variants [43,51]. This finding suggested that the current dogma that a significant percentage of EHM outbreaks are caused by a mutant strain (G2254) is too overly simplistic [52].

Diagnosis: Over the past 15 years, diagnosis tools have been improved (Table 1). PCR that allows for the direct detection of the virus has become the new standard [41]. The use of this powerful diagnostic assay has to nevertheless be considered with all the clinical information available by the practitioner, in particular the time of completion of the sampling (nasal swab and blood). Indeed, the viral load observed during EHM is generally much lower than the one observed during abortions. Given that EHV-1 latently infects leukocytes, PCR results obtained from blood samples should be interpreted with care. It is important for samples taken a few days after the observation of neurological clinical signs, not to exclude the possibility of an infection with EHV-1—even when a negative PCR result was obtained. Monitoring contact horses with PCR tests performed on nasopharyngeal swabs is then recommended. The discriminatory test between the neuropathogenic and non-neuropathogenic strains is also used in many laboratories (SNP PCR A/G2254), but regardless of the results of the SNP PCR, the practitioners will have to apply equivalent EHV-1 management measures. Virus isolation, serological testing (virus neutralization) and post-mortem examination are still informative.

Prevention and control: Practitioners need to identify quickly EHV-1 infections and to apply strict sanitary measures to stop virus spreading. Isolation and quarantine measures have to be applied according to high-risk groups. There is an urgent need to screen and separate potential virus shedders (either confirmed to be infected with EHV-1 or exposed) from non-exposed and healthy animals. Specific staff caregivers would be affected to each group and would have specific supplies (gloves, coats, calots, boots and footbath). A three weeks’ quarantine starts as soon as the last reported case is declared. At the end of the infected period, an eight days’ crawlspace, after carrying out cleaning and disinfection of boxes, is needed.

Several inactivated and live vaccines are available against EHV-1 and EHV-4, and both types of vaccines have been marketed in Europe (Table 2) (reviewed in [53]). Although they reduce both clinical signs of the respiratory disease and virus shedding [54], their efficacy against neurological disorders and abortion is limited [41]. In respect to these limitations, practitioners have turned toward alternative treatments by using antiviral molecules—even if no marketing authorization are available in horses. In vivo, valaciclovir, which is the prodrug of aciclovir, was tested in experimental EHV-1 infection, but showed no antiviral effects [55]. *In vitro* studies are performed around the world to identify new molecules with a strong antiviral potential [56,57].

### 2.2. Rabies Virus

Rabies virus is a neurotropic virus belonging to the genus *Lyssavirus*, family *Rhabdoviridae.* It is responsible for a zoonotic and inevitably fatal disease, once neurological signs have been recognized and is nevertheless considered as a neglected disease in tropical areas. Every mammal is susceptible, but some species such as dogs, jackals, coyotes, wolves, foxes, skunks, mongooses, raccoons, and bats act as reservoir hosts.

Virus: Lyssaviruses are currently classified into 17 different species: *Rabies virus* (RABV), *Lagos bat virus* (LBV), *Mokola virus* (MOKV), *Duvenhage virus* (DUVV), *European bat lyssavirus types 1* and *2* (EBLV-1 and -2), *Australian bat lyssavirus* (ABLV), *Aravan virus* (ARAV), *Khujand virus* (KHUV), *Irkut virus* (IRKV), *West Caucasian bat virus* (WCBV), *Shimoni bat virus* (SHIBV), and more recently described bat lyssaviruses (*Bokeloh bat lyssavirus* (BBLV), *Ikoma lyssavirus* (IKOV), *Gannoruwa bat lyssavirus* (GBLV) and *Lleida bat lyssavirus* (LLEBV)) [6,58,59,60,61,62,63]. Lyssaviruses are also separated into three phylogroups, based on their genetic, immunologic, and pathogenic characteristics. Phylogroup I includes RABV, DUVV, EBLV-1, EBLV-2, ABLV, ARAV, IRKV, BBLV, GBLV, and KHUV, phylogroup II includes LBV, MOKV, and SHIBV, and phylogroup III includes WCBV, IKOV, and LLEBV [59]. All lyssaviruses are capable of causing fatal acute encephalitis indistinguishable from clinical rabies in humans and other mammals. With the exception of Mokola and Ikoma lyssaviruses, every species have known bat reservoirs, leading to the speculation that lyssaviruses originated in the order Chiroptera [62]. Human clinical rabies cases have been documented for RABV, MOKV, DUVV, EBLV-1, EBLV-2, ABLV, and IRKV [63].

Lyssaviruses are enveloped, bullet-shaped viruses with a single-stranded, negative sense RNA genome of about 12 kb that encodes five viral proteins: nucleoprotein (N), phosphoprotein (P), matrix (M), glycoprotein (G), and RNA polymerase (L) (Figure 1). The RNA genome is encapsidated by the N protein, forming the ribonucleoprotein (RNP) complex, which is the functional template for transcription and replication.

Transmission and epidemiology: Rabies cases have been reported across the globe in more than 150 countries [5]. According to recent estimates by the World Health Organization, 55,000 to 60,000 human deaths due to rabies infection are expected to occur every year [64]. The majority of them occur in developing countries in Asia and Africa, with about 35,000 and 21,000 human cases, respectively, and rabies virus is usually transmitted by free roaming dogs in these areas [64,65]. Two major epidemiological cycles are reported, urban canine rabies, now largely confined to developing countries and sylvatic or wildlife rabies which predominates throughout most of Europe and North America [65]. Animal species involved in rabies virus transmission along sylvatic cycles may vary. In the United States, skunks, raccoons and bats are the wild species most often found rabid, while in Canada, these are foxes and skunks. In Europe, the red foxes and raccoon dogs serve as the main reservoir hosts and red foxes were found to account for 60% of all reported cases in central and western Europe. Bat species involved in rabies virus transmission also differ between countries: in Latin America, vampire bats are an exceptionally devastating source of infection for cattle and equids, while in North America rabid non-hematophagous bats have occasionally transmitted the disease to horses [66,67]. Horses are sensitive to both canine and bat rabies strains. Rabies is fairly rare in horses and usually less than 100 cases are reported in the United States every year: horses and mules (*Equus* spp.; 31 [6.6% of rabid animals] in 2013) [68] (Figure 5). A large number of rabies cases (172 out of 467 suspected cases) have been reported in donkeys in Sudan over a period of 10 years from 1992 to 2002 [69]. In Australia, two equine cases also arose recently in 2013, documenting the first occurrence of ABLV in animals other than bats or humans [70]. In Europe, 233 cases have been reported between 2010 and 2019, mainly in Eastern Europe (Russian Federation (49), Ukraine (41), Turkey (70), Belarus (27), Moldova (9), Romania (17), Georgia (8), Poland (2), Croatia (7), Serbia (1), and Latvia (1)), while for the same period, only one case was reported in Western Europe, in Italy (2010) [71].

Pathogenesis and clinical disease: Rabies virus is primarily transmitted to equids through the saliva of an infected animal. Contamination occurs mainly through bites or contact of a cutaneous or mucous (oral, nasal, eye mucosa) lesion with infected saliva. Rabies pathogenesis is characterized by three distinct phases. Phase 1 corresponds to the ascending or centripetal period during which the virus is transported toward the CNS. Phase 1 occurs after the bite of a rabid animal, and after a short-lasting replication in local muscle cells, the virus enters motor and sensory neurons. Paresthesia at the biting site may develop, which results in rubbing or automutilation through biting. Lyssavirus mainly shows axon-neuronal transport by binding with acetylcholine-receptors at motor end plate and multiply at the ventral horn of the spinal cord before CNS spreading [72]. The virus replicates within the CNS during phase 2, leading to clinical signs of encephalomyelitis. Phase 2 in horses is characterized by extensive virus replication in the limbic system (including hypothalamus, hippocampus, amygdala, and other nearby areas) and the spinal cord [73]. Because phase 2 is the period with the most dramatic clinical signs, most horses will be euthanized during that phase. Phase 3, also called centrifugal phase, is the period where the virus leaves the CNS and infects other organs in the body. Phase 3 is characterized by neuronal transportation of virus into highly vascularized organs, such as the salivary glands, facilitating virus transmission to new hosts and its excretion into the environment.

Clinical signs of rabies are highly variable in horses and three forms are classically described according to the injured area: silent, paralytic or furious forms. The furious form is uncommon (10 to 17% of cases), while silent and paralytic forms are the most common. After a long incubation time (over 6 months), the disease progresses rapidly (within 3 to 6 days), including a sudden change in behaviour (depression to manic), itching at the biting site, loss of appetite, high fever, gait disorder, paralysis at the inoculation point, aggressivity, and hyperesthesia [74].

Diagnosis: Confirmatory diagnosis is preferably undertaken through virus identification by direct fluorescent antibody (DFA) test, direct rapid immunohistochemistry test (dRIT), or pan-lyssavirus RT-PCR assays [75]. DFA test, dRIT, and RT-PCR provide a reliable diagnosis in 98% to 100% of cases for all lyssavirus strains if an appropriate conjugate or primer/probe is used [76]. For a large number of samples, conventional and real-time PCR can provide rapid results in equipped laboratories. Histological techniques such as Seller staining (evidencing Negri bodies) are no longer recommended for diagnosis. In case of inconclusive results from primary diagnosis tests (DFA test, dRIT, or pan-Lyssavirus RT-PCR), further confirmatory tests (molecular tests, cell culture or mouse inoculation tests) on the same sample or repeated tests on additional samples are recommended. Wherever possible, virus isolation in cell culture should replace mouse inoculation tests.

Prevention and control: Rabies control has been mainly afforded through the vaccination of wild and domestic susceptible animal species [77]. Vaccination is recommended in endemic areas. Several inactivated adjuvant vaccines are commercialized in Europe and can be used in domestic mammals (see Table 2 for the list). Rabies vaccination intervals >1 year may be appropriate for previously vaccinated horses, but not in primed horses vaccinated only once [78].

### 2.3. Borna Disease Virus

The Borna disease virus (BDV, renamed BoDV), *Mammalian 1 orthobornavirus* according to ICTV nomenclature [6], is the prototype member of the *Bornaviridae* family, within the order *Mononegavirales* [79]. For years, it was the only member of this family, but since 2008 new bornaviruses were discovered in birds, reptiles, and mammals (reviewed in [80]). Amongst them, an avian borna virus (ABV), the Psittaciform 1 orthobornavirus, was shown to be responsible for the proventricular dilatation disease [81], and a mammal Bornavirus, the Variegated squirrel 1 bornavirus (VSBV-1), recently reappointed *Mammalian 2 orthobornavirus*, was associated with fatal encephalitis in humans [82]. This expansion of the *Bornaviridae* family and the association of the new viruses with animal and human diseases revived the interest for this family and called for new classification [80].

Virus: Bornaviruses are enveloped, non-segmented, single-stranded, negative-sense RNA viruses. Their 8.9 kb genome encodes six viral proteins, five structural (nucleoprotein N, phosphoprotein P, matrix M, surface glycoprotein G and the large structural protein L directing the replication of BoDV RNA genome) and one non-structural (X) (Figure 1). They have the particularity, amongst the Mononegavirales, to replicate within the nucleus and to be poorly released from infected cells [83].

Transmission and epidemiology: BoDV is the causative agent of the Borna disease, a rare but severe, often lethal, encephalitis, which was first described in horses during a devastating outbreak in the little city of Borna in Saxony/Germany around 1894 [84]. Later, it was reported to infect a large range of animals, including sheep, cattle, dogs, cats, shrews, ostriches, birds, macaques, and several zoo animals, suggesting that its potential host-range includes all warm-blooded animals [85,86,87,88,89,90,91]. BoDV infection, based on antibodies, antigen, RNA and virus detection, has been reported from horses and other animals in many countries in several continents, Europe, North America, Australia, and Asia, suggesting a worldwide distribution of the virus (reviewed in [89]). This should, however, be taken cautiously as diagnosis is not always reliable (see Diagnosis section below). Illustrating these difficulties, viral RNA detection outside central Europe, the endemic area in which the highest clinical incidence was consistently found as well as the verified classical Borna disease’s cases, has been suspected to be caused by contamination [92]. Endemic area includes eastern and southern Germany, the eastern part of Switzerland and the area bordering Liechtenstein as well as the most western part of Austria (reviewed in [92,93]) and, as recently reported, upper Austria [94] (Figure 6).

The bicolored white-toothed shrew, *Crocidura leucodon*, is recognized as the natural reservoir host of BoDV [88,94,95]. In this host, the virus replicates in numerous tissues, neural and non-neural, without causing clinical symptoms or pathological lesions. It is secreted in saliva, urine, skin, tears, and feces [96]. Horses may be infected via the olfactory route, as the presence of BoDV antigen and RNA, as well as inflammation and edema, have been found in the olfactory bulb of naturally infected horses early in the course of the disease [97]. This is supported by successful experimental intranasal infection of rats, mice, sheep, and horses (reviewed in [93]). In horses, the virus is mostly confined into the brain and, although BoDV RNA was found in oral, nasal and conjunctival fluids of naturally infected horses, infectious virus was rarely detected [98], indicating that transmission from horse to horse in stables is unlikely. Vertical transmission may be possible as viral RNA was detected in the brain of a pregnant mare and her fetus [99] and was shown to occur experimentally in mice [100].

The question of whether BoDV is a human pathogen has been debated for years. Several studies showed it may be a cause for some psychiatric disorders [101,102], and possible responsible mechanisms have been proposed [103,104], but others have suspected that contamination occurred in initial studies [105], and have refuted any association between BoDV infection and mental illness [106]. While this remains a question, two recent studies convincingly showed that BoDV is a human pathogen, as it was associated with fatal encephalitis [107,108]. How humans have been infected remains to be elucidated but the proximity of BoDV sequences from humans, shrews and horses leads to suspect a zoonotic risk.

Pathogenesis and clinical disease: In horses, the typical course of the disease is characterized by an acute encephalitis that develops following an incubation period which lasts up to 3 months [109]. During the acute phase, neurological and neuro-behavioural signs vary but may include unusual posture (crossing legs), repetitive movement disturbance, teeth grinding, circle walking, neck stiffness, nystagmus, strabismus, myosis associated with external stimuli such as hyper excitability, aggressivity, lethargy, sleepiness and stupor. Hyperthermia, which may precede neurological signs, is not always noticed during Borna disease. During the final stage, paralysis may appear followed by seizures associated with specific movements named “push to the wall”. Death occurs in 80% to 100% of cases, in 1 to 4 weeks after the onset of clinical signs. In survivors, infection is life-long, and a chronic form of the disease develops with recurrent clinical events such as depression, apathy, somnolence and scared behavior, in particular following stress [110]. Of note, some infection may be asymptomatic. Histopathological examination of infected brains revealed viral antigens mainly in neuronal nuclei and the characteristic Joest-Degen inclusion bodies, accompanied with massive infiltration of inflammatory cells [109,110].

Diagnosis: The diagnostic of BoDV infection is particularly difficult. Clinical signs are not specific and low titres of antibodies in infected horses and low viremia (the virus is confined within the brain) does not allow a reliable diagnostic from serum or cerebrospinal fluids, even when the most sensitive serological or molecular tests are used. Standardized tests, validated by inter-laboratory assays, does not exist. Although intra-vitam studies give useful indicators, only *post-mortem* analyses performed in brain, the tissue with the highest viral load, will confirm a definitive diagnostic (Table 1).

Prevention and control: A few therapeutics (amantadine sulfate) and vaccines (attenuated and inactivated candidates) have been developed against equine BoDV infection thus far, but none are available in veterinary medicine since none proved effective in controlling or preventing the disease [109].

### 2.4. Enzootic Flaviviruses: West Nile Virus, Tick-Borne Encephalitis Virus and Louping Ill Virus

Three neurotropic flaviviruses documented in horses suffering from meningoencephalitis are enzootic in Europe, West Nile virus (WNV), tick-borne encephalitis virus (TBEV) and Louping ill virus (LIV). Flaviviruses can be divided into three distinct groups according to their vectors: tick-borne viruses, mosquito-borne viruses and viruses with unknown vectors [111].

Viruses: Flaviviruses, belonging to the *Flaviviridae* family, are enveloped, non-segmented, single-stranded and positive-sense RNA viruses. Their genome of approximately 11 kb encodes three structural proteins (capsid C, preMembrane prM and Envelop E) and seven non-structural proteins (NS1, NS2A, NS2B, NS3, NS4A, NS4B, and NS5) involved in virus replication and counteractive of immune responses (Figure 1). Notably, flaviviruses belonging to the Japanese Encephalitis serocomplex (such as WNV, Japanese, Saint-Louis and Murray Valley encephalitis viruses) express an additional non-structural protein, NS1′, resulting from ribosomal frameshift occurring at a specific heptanucleotide motif close to the beginning of the NS2A gene [112]. Even though the precise functions of NS1′ are still largely unknown, this protein has been involved in virus neuroinvasiveness.

#### Transmission and Epidemiology

##### WNV

WNV is maintained and amplified in an enzootic cycle involving birds and mosquitoes from the *Culex* genus as vectors. WNV is transmitted to different animal species (mainly mammals but also reptiles and amphibians) through the bite of infected mosquitoes. Horses and humans are highly susceptible to WNV infection but are considered as dead-end hosts owing to limited and short viremia that does not sustain transmission to naïve mosquitoes. In both species, asymptomatic infections are the most common, but in rare cases (approximately 1 out of 140 infections in humans and up to 10% infected horses), neuroinvasive forms with meningitis, encephalitis or myelitis may occur [113,114]. WNV is the most widely distributed arbovirus that induces equine encephalitis (Figure 7). Over the last 15 years, WNV has been repeatedly reported in Europe with a high frequency in the Mediterranean region and in Eastern Europe. This virus was first described in France, Portugal and Cyprus in the 1960s [115]. After a silence of more than 30 years, WNV lineage 1 strains resurfaced in North Africa (in Morocco (1996), Algeria (1994) and Tunisia (1997)), as well as in Western and Eastern Europe (Romania (1996), Italy (1998), Russia (1999) and France (2000)) [116,117,118]. WNV strikingly exemplified how fast and unpredictable flaviviruses can emerge when the virus was introduced in New York City in 1999. It produced large and dramatic outbreaks in humans and horses and rapidly spread, in less than 4 years, throughout the United States of America, causing more than 30,000 cases and 1200 deaths in humans and more than 24,000 cases in the equine population for the United States only over a 10-year period [119]. Interestingly, in Europe the epidemiological scenario in 1996–2010 was quite different from the one in North America as epidemics were irregular and limited in time and space. Nevertheless, a revival of WNV activity in Europe has been associated in particular with the introduction in 2004 of a new WNV strain within lineage 2, most likely originating from Africa [120]. This WNV lineage 2 was initially identified in Hungary and then spread to the eastern part of Austria and to southern European countries including Greece in 2010 and Italy in 2011 [121]. Unprecedented WNV transmission seasons in Europe were registered in 2010, 2012, 2013, or 2015, in association with climatic and environmental conditions sustaining mosquito activity and close mosquito-bird contact rates. Nevertheless, these recent transmission seasons were in no way comparable to the exceptional transmission wave experienced in 2018. Indeed, 2018 showed a 7.2-fold increase of reported cases compared to the 2017 transmission season and a final total number of reported autochthonous infections in humans (*n* = 2083) higher than the cumulative number from the previous seven years (*n* = 1832) [17]. The highest increase compared to 2017 was observed in Bulgaria (15-fold) followed by France (13.5-fold) and Italy (10.9-fold). The number of European equine WNV outbreaks doubled in 2018 (*n* = 285) in comparison with earlier WNV transmission seasons (*n* = 97–191 in 2013–2017, with on average 145 equine cases reported annually for this period) [17]. The second remarkable pattern was the reporting in 2018 for the first time of WNV in northern Europe, with several bird species and two horses found infected by WNV lineage 2 in Germany [122]. In Australia, specific virus variants called Kunjin virus and classified into WNV lineage 1b, recently caused unprecedented epizootics of neurological disease in horses in Southeast Australia, resulting in almost 1000 cases and a 9% case fatality rate in 2011; unusual climatic conditions, as well as enhanced virus transmission by infected mosquitoes, could have contributed to the phenomenon [123].

WNV neurovirulence and neuroinvasion are typically associated with sequence variations in the flavivirus E protein [124]; strikingly, a unique mutation at position 249 in the helicase portion of NS3 (NS3249P) has been identified in virus strains that have been responsible for major WNV outbreaks during the two last decades [125] and its role in the modulation of WNV virulence and transmission is nowadays largely debated [126,127].

##### TBEV and LIV

TBEV is the most important human tick-borne pathogen in Europe and Asia. The estimated annual incidence rate is 10,000 human infections with a case-fatality rate ranging from 1% to 20%. For TBEV, the arthropod vectors are primarily hard ticks and in Europe, the most important tick vector is *Ixodes ricinus*. Contrary to mosquitoes, which become infected only if there is sufficient viremia in the vertebrate host, ticks can become infected during a shared meal or “co-feeding”, not requiring a systemic infection of the host [128]. In fact, adult and immature ticks, as well as larvae and nymphs are attached to their host for several days and can feed together on the same reservoir. Contamination between naïve (uninfected) and infected ticks can occur during this meal. Co-feeding is facilitated by the proximity of ticks and the action of saliva, which allows for the transfer of arboviruses including TBEV [129]. The main vertebrate reservoir hosts of TBEV are rodents of the genus *Myodes* and *Apodemus* although other small rodents and shrews can contribute to the natural transmission cycle. Larger animals such as sheep, goats and more rarely cattle can be additional competent hosts. Goats, sheep and cows excrete the virus in the milk. Humans, horses and game (deer, wild boar, fox) are epidemiological dead-end hosts. Human infection can occur through a bite from a TBEV- infected tick, more rarely through the ingestion of unpasteurized milk or milk products from goats and less often from infected cows or sheep [130]. Hard ticks (*Ixodes ricinus*) also transmit LIV. The vertebrate reservoir hosts are the wood mouse (*Apodemus sylvaticus*), the common shrew (*Sorex araneus*), the red grouse (*Lagopus lagopus scoticus*) and the sheep (*Ovis aries*). A co-feeding mechanism has also been reported for LIV transmission in *I. ricinus* ticks feeding on mountain hares (*Lepus timidus*) [131]. Sheep develop the disease and the virus could occasionally been detected in a range of other animal species such as goats, dogs, pigs, horses, humans, deer, llamas, alpacas, and mountain hare [132]. Finally, very uncommon cases of horse infected by TBEV or LIV have been described [133,134,135] and a few serological surveys in equids are available in the scientific literature. TBEV seroprevalence rates of 20% to 30% among asymptomatic horses have been reported in Austria and Germany [134,136] while lower ones have been reported in the Balkans (3% to 5% in Serbia and Slovakia) [137].

TBEV is reported in the northern hemisphere of Europe and Asia. There are three main subtypes of TBEV: European (TBEV-Eu), Siberian (TBEV-Si) and Far Eastern (TBEV-FE) circulating in Europe with TBEV-Si and TBEV-FE recently detected in the Baltic countries and in Eastern Finland [138,139]. Based on epidemiological investigations, LIV distribution area initially limited to the British Isles (particularly in Scotland, Cumbria, Wales, Devon and Ireland), but seroconversion or clinical cases in sheep due to LIV-like viruses have been reported during the last decade in Norway, Denmark, and Spain [140,141].

Pathogenesis and clinical disease: TBEV, LIV and WNV induce severe neurological syndromes, through pathogenic mechanisms that are still largely unknown. The majority of WNV and TBEV studies were done in vitro on transformed cells lines and in vivo on mouse experimental models [142]. The exact mechanism of WNV and TBEV CNS invasion is unclear, but five models have been proposed that rely on the anatomy of the blood-brain barrier (BBB): (i) entry through the BBB via infected leukocytes (a so-called Trojan horse mechanism, demonstrated for WNV) [143]; (ii) a direct passage of the hemato meningeal barrier after its integrity has been compromised by the action of cytokines (TNFα) or metalloproteinases (MMP9) inducing changes in capillary permeability (WNV) [144]; (iii) direct infection of the brain microvascular endothelial cells without effect on cells integrity (WNV and TBEV) [145]; (iiii) infection or passive transport through the epithelial cells of choroid plexuses, whose function is the production of CSF (WNV) [146]; (iiiii) infection via the neuronal pathway by the infection of olfactory neurons and/or axonal transport in the retrograde direction (mechanisms demonstrated for WNV only) [147]. This latter axonal transport promotes entry into the CNS from peripheral inoculation near nerve connections and acute flaccid paralysis of a limb. In an experimental model of infection of hamster, WNV was present 4–5 days after infection in multiple sites of the brain and spinal cord. Foci of neuronal infection were observed in the cortex, hippocampus, cerebellum, basal ganglia or the anterior horn of the spinal cord [148]. Neurons and astrocytes are infected with WNV and TBEV [142,149,150]. Following neuronal cell death, inflammatory molecules (such as IL1-β, IL6, IL8 and TNFα) have potentially toxic effects on uninfected neurons. Later, during infection, lymphoid infiltration can be observed in the infected regions with cells releasing proinflammatory cytokines destroying flavivirus-infected cells but also contributing to the pathogenesis of the virus by their cytotoxic action in the CNS [133,151].

An equine infection by WNV can be suspected when the following symptoms are noticed: hyperthermia, ataxia, hind legs paresis, muscle tremors, teeth grinding, cranial nerve deficits, dysphagia or face paralysis [9]. During 2000 and 2004 WNV epizootics in Camargue, France, ataxia was the main clinical sign observed in 64% to 72% of cases, while behavioral modifications (45% of cases) and muscle tremors (35% of cases) were less frequently reported [9]. Neurological disorders could persist within 5 to 30 days with a complete or partial remission after several months. Case fatality rates are variable, generally ranging from 20% to 57%.

Few publications describe TBEV infections in animals. Clinical signs of encephalitis induced by TBEV are rare in horses and include anxiety, decreased appetite, nervousness, and emaciation [134,135]. Generally asymptomatic in horses, the confirmation of equine TBEV infections can be challenging due to close and partially shared antigenicity with other flaviviruses such as WNV and Japanese Encephalitis Virus (JEV) [136]. Encephalitis caused by LIV in horses is very uncommon. During the outbreak reported in Ireland, horses developed neurological disorders comparable to the ones described for other arborviral infections affecting the CNS. Its principal symptoms consisted of ataxia ranging from slight incoordination to falling risk. Face and neck muscle tremors, depression, fear of light, behavioral modification such as constant chewing or mild fever could be observed [133,152].

Diagnosis: The neurological symptoms and lesions are not specific to flavivirus infection, making laboratory tests compulsory to confirm or exclude viral etiology. The diagnosis of flavivirus infection is delicate and relies on the detection of viral RNA in blood and CSF, viral isolation in cell culture and/or the detection of IgM and IgG in serum and CSF. Flavivirus infection (WNV, TBEV, LIV) diagnosis is primarily based on indirect methods. Indeed, viremia of an infected horse is quite low and fleeting and already vanished by the time horses develop neurological signs. It is therefore quite difficult to identify the pathogen by direct diagnostic assays, even with highly sensitive real-time RT-PCR assays. Serology tests such as ELISA for WNV and TBEV detection and hemagglutination inhibition assay for LIV are fast and allow for the identification of anti-flavivirus antibodies. These tests suffer from poor specificity because of antibody cross-reactions between flaviviruses [153]. Several serological tests have to be carried out in order to detect a recent infection (IgM ELISA for WNV or an antibody titer kinetics). A virus neutralisation test, with higher diagnostic specificity, practiced in a biological security 3 level laboratory allows the definitive identification of the flavivirus. WNV infection can also be diagnosed by RT-PCR on EDTA-blood samples collected during the first clinical period (marked by hyperthermia only, a few days after mosquito bites and before the neuro-invasive form of the disease) or on cerebrospinal fluid and brain on post-mortem. Samples must be shipped cold or frozen, as WNV is very sensitive to thermal and chemical inactivation.

Prevention and control: There is no specific treatment for flaviviruses and disease control primarily relies on integrated virus surveillance and on vector control, with strategies and methodologies differing between European countries [154,155]. However, for WNV, three equine vaccines have a European Marketing Authorization (Table 2). The Zoetis Equip WNV vaccine is composed of the inactivated West Nile virus, strain VM-2 New York 1999 combined with an adjuvant [156]. The other two vaccines are recombinant and adjuvanted vaccines. Proteq West Nile is composed of a canary poxvirus vector expressing the prM and E genes of WNV [157]. The canarypox vector performs an abortive replication cycle in mammalian cells where the inserted gene product (transgene) is expressed [158]. Finally, the Equilis West Nile vaccine from Intervet consists of the yellow fever virus (YFV) 17D vaccine strain where the prM and E genes of YFV have been replaced by those of WNV. This recombinant vaccine is injected under an inactivated form [159]. No vaccine against TBEV and LIV are available for horses.

## 3. Exotic Equine Encephalitis Viruses in Europe

Equine encephalitis viruses enzootic in Europe are not the only threat to European horses. Horses could contract the disease abroad during equine competitions or international events. Moreover, because of enhanced risks of emergence of exotic viruses and increasing animal movements owing to globalization, exotic equine encephalitis viruses could be reported in the future in new naïve territories. Expert opinion and risk assessment through modelling is strongly required to identify the viruses that are prone to emergence and the most at-risk areas in Europe and at the international level and to adapt surveillance plans [160,161].

The main exotic equine encephalitis viruses are vector-borne, either transmitted by mosquitoes or flying midges (Figure 3b). They encompass viruses belonging to three genera, flaviviruses in the *Flaviviridae* family (Japanese encephalitis virus (JEV), Saint-Louis encephalitis virus (SLEV) and Murray Valley encephalitis virus (MVEV)), alphaviruses in the *Togaviridae* family (eastern, western and venezuelan equine encephalitis viruses, EEEV, WEEV and VEEV respectively) and orbiviruses in the *Reoviridae* family (equine encephalosis virus, EEV) [21]. Most of them, with the exception of EEV, are zoonotic. During such zoonotic infections, humans and equines will generally not develop viremia elevated enough to infect naïve mosquitoes and they are considered as dead end hosts; horses can however effectively replicate epizootic strains of VEEV, I-AB and I-C variants, developing high viremia for an average of 4 days and serving as virus reservoir for mosquito transmission to animals or humans [20,21].

Pertaining to flavivirus infections, JEV, SLEV and MVEV induce similar clinical presentations to the ones reported with WNV but mainly differ in their case fatality rate and more restricted host and vector ranges, likely contributing to more limited geographical range (reviewed in [12,20,162,163]). Case fatality rates reported for JEV, SLEV and MVEV are 5% to 30%, 3% to 30%, and 15% to 20%, respectively [5,164]. JEV, SLEV and MVEV have been mainly reported from South-Eastern Asia, America (from Northern America up to Argentina) and Northern Australia/Papua New Guinea respectively (Figure 8). JEV and SLEV infections have been unfrequently reported in horses during the last decade, while MVEV equine outbreaks across south-eastern Australia have been identified in 2011 [12]. However, among these exotic flaviviruses, JEV presented the highest propensity to spread, with transmission evidenced westward to Nepal and Pakistan, as well as eastward in western Pacific regions in the 1990s (Eastern-most territories in Indonesia, Papua New Guinea and Australia in 1995 and 1998), subsequent to changes in human activities (deforestation, irrigation, expansion of pig breeding) [165]. Intriguingly, JEV genome fragments have been recently identified in birds and mosquitoes collected in Italy [166,167]. JEV actively circulates in rice paddies, rural and semi-urban areas, amplified by Ardeid birds (herons and egrets) and pigs as natural maintenance reservoirs and *Culex tritaeniorhynchus* or other mosquito species mainly from the *Culex* genus as vectors (reviewed in [20]). Finally, SLEV appears as a recently re-emerging flavivirus. Subsequent to WNV introduction in Northern America in 1999, probable virus competition for avian amplifier hosts likely contributed to initial disappearance of SLEV from the Western United States (1999–2014) [168,169]. From 2015, SLEV has been again reported in California and the re-emerging strain clusters genetically with an epidemic strain identified 10 years earlier during an unprecedented human encephalitis outbreak in Cordoba, Argentina [162,170].

EEEV, WEEV, and VEEV are alphaviruses regularly identified in equine encephalitis in America. These enveloped, non-segmented, single-stranded and positive-sense RNA viruses encode two polyprotein gene clusters, driving the translation of four non-structural proteins (NSP1-4) and virus structural proteins (and in particular two Envelop proteins, E1 and E2) at the 5′ and 3′ ends, respectively (Figure 1). Epidemics or epizootics of EEE, WEE, and VEE have been recognized at irregular intervals since 1831, 1847, and 1939, respectively, in different regions of America (Figure 8) [171]. The last documented human WEE case in North America occurred in 1994 and the virus has not been detected in mosquito pools since 2008, while severe epizootics involving more than 41 horses have been identified in 2019 in Mexico (Promed archive 20190406.6407111) [16,172]. The most severe forms of alphavirus equine encephalitis are associated with EEEV and VEEV epizootic variants. VEEV strains are classified into six subtypes (subtypes I-VI), with subtype I including five antigenic variants (A, B, C, D, E, and F) and epizootic strains corresponding to I-AB and C strains only. All other subtypes and other subtype I variants are endemic strains and seldom cause encephalitis in horses. Epizootic I-AB and C strains were shown to arise from E2 mutations of enzootic I-D or E variants leading to increased protein positive charges and in particular from T213R/K substitution [173] and have been described in Argentina, Peru, Columbia, Ecuador, Mexico, Trinidad, Venezuela, the United States, as well as in Panama in 2019 [16,174]. VEEV enzootic strains are mostly maintained in birds and rodents belonging to *Sigmodon*, *Zygodontomys*, *Heteromys* and *Proechimys* genus and in *Culex melanoconion* mosquitoes, while epizootic variants are transmitted by more diverse mosquito species (*Aedes*, *Psorophora* genus) [175]. Regarding EEEV, North and South American variants have been described, with North American variants being the most pathogenic for mammals [176,177]. Alternate infection of birds and *Culiseta melanura* or *morsitans* mosquitoes maintain this virus in nature, while *Aedes* bridge vectors are involved in EEEV transmission to humans and horses [20]. EEEV clinical infections in horses are highly lethal, with 70% to 90% case fatality rate reported in the literature, in comparison with 20% to 50% for WEEV and 40% to 80% for VEEV (reviewed in [20]). EEEV equine epizootics are regularly observed in the United States and in Canada, Ontario and in this respect, 2019 has sustained active transmission in North America, with more than 18 states from Eastern United States (the western limit being delineated by Minnesota and Louisiana) having reported more than 99 EEEV horse cases from early spring (end of March-April) to late autumn [16].

Equine encephalosis virus (EEV) is an orbivirus close to African horse sickness and bluetongue viruses, two arboviruses associated with unexpected emergence in Europe in 1989 and 2006 respectively [178,179]. It is a primary endotheliotropic virus and most EEV infection cases are asymptomatic or poorly symptomatic, but its clinical presentation may include ataxia, depression, hyper excitability and convulsions. Since its description in Israel in 2008–2009, it is of primary importance to improve the preparedness of European countries to EEV emergence [180].

## 4. Conclusions

Many neurological diseases represent an important sanitary and economic threat to the horse population worldwide. Non-vector-borne equine encephalitis viruses, such as EHV-1, rabies and BoDV, are variably reported in Europe, whereas arthropod-borne infections are usually exotic diseases. Viral encephalitis therapeutic strategies are comparable, whatever the etiologic virus. Horses with neurological conditions must be isolated in a quiet box with limited stimuli (noise, light) and an appropriate bedding providing warmth, comfort, and security. Slings can be used to support paretic horses and avoid long and poor prognosis recumbency (Figure 2). Supportive care will contribute substantially, avoiding complications and improving the prognosis. The use of DMSO (0.4–0.9 g/kg for 5–6 days) has been advocated on the basis of its free radical scavenging properties but its efficacy has not been evaluated scientifically [1]. Nonsteroidal anti-inflammatory drugs may be used to control pyrexia, inflammation and discomfort, while short-term use of glucocorticoids may be beneficial; glucocorticoids proved to be valuable in some EHM horses and it is hypothesized that the treatment reduces the supposed immune-mediated EHV-1 pathogenesis [181]. However, they were also shown to reactivate latent herpesvirus infection and to increase the level and duration of virus shedding [182]. Vaccination is controversial in the face of outbreaks and in particular inactivated vaccines take too long to generate immune responses capable to limit disease spread when outbreaks are seasonal (WNV, other vector-borne viruses in temperate areas).

Clinical signs of viral equine encephalitis are not specific and overlapping geographical areas can make virus identification very challenging. One recent and striking example of delayed identification of emerging arboviruses due to similarities in clinical presentation and cross-reactive diagnostic tools was given during WNV introduction in the United States, when WNV was initially misdiagnosed with the closely related SLEV [183]. Bearing in mind that three flaviviruses responsible for equine encephalitis are described in Europe, and that serological cross-reactivity is frequently observed in flavivirus indirect diagnosis assays, the development of multiplex approaches that allow the comparison of serological reactions against a wide range of pathogens appear to be valuable options [184,185]. Furthermore, because in about one-half of infectious equine encephalitis, no known pathogen can be evidenced [4], identification of unknown neuropathogenic viruses by classical (electron microscopy) and more recent high-throughput techniques (next generation sequencing for example) is highly desirable [186,187]. In these two recent studies, three viruses, Shuni virus, horse parvovirus-CSF and eqcopivirus, have been identified as potential causes of neurologic disease in horses through unbiased detection from different tissues or body fluids; the demonstration of infectious virus from the brain of sick horses establish Shuni virus as a novel equine neuropathogenic virus [186], while for the other two viruses for which genomic DNA was detected in CSF and/or plasma [187], comparison of virus prevalence in the CSF of healthy horses (case-control study) would be required before a conclusion on the aetiology of equine encephalitis can be reached.

Arboviruses are the most important cause of encephalitis in horses and many of these viruses are also significant human pathogens. Some of these arboviruses have recently emerged or resurged, such as WNV, JEV, SLEV, EEEV or EEV and an increased rate of emergence of vector-borne diseases can be inferred from recent studies [188]. A high diversity of mosquito species have been reported in Europe (mainly from *Aedes*, *Culex* and *Culiseta* genera), and highly invasive *Aedes albopictus* and *Ae. japonicus* have rapidly established in several European countries over the last decade [189,190]. Vector competence of native and invasive European mosquito species for equine encephalitis viruses, other than WNV and JEV, has been unfrequently evaluated [191,192,193]. Consequently, identification of European regions at risk for the spread of exotic equine encephalitis viruses is difficult and mainly relies on information on mosquito and animal hosts density and on records of opportunistic mosquito species [160]. On-time control of vector-borne infections relies on the use of sentinel systems, including horses or sentinel chicken flocks for example, to provide warning of virus activity and initiate mosquito control measures [155].

## Figures and Tables

**Figure 1 viruses-12-00023-f001:**
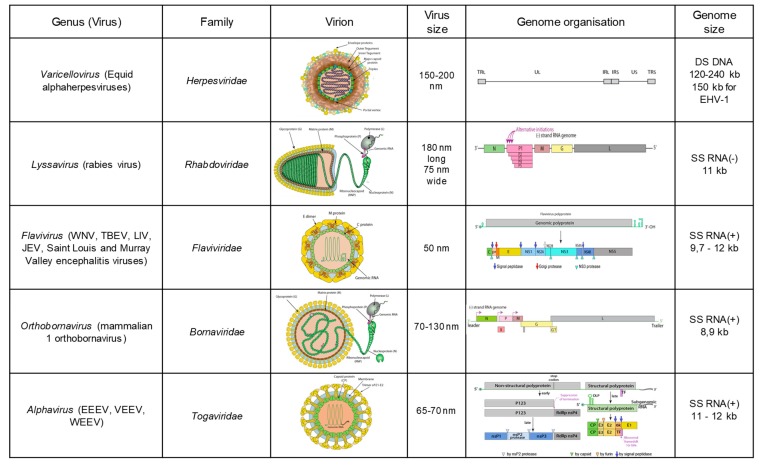
Major viruses causing encephalitis in equines. Virus classification according to ICTV 2019 nomenclature [6], structure and genome organisation are presented for viruses belonging to *Herpesviridae*, *Rhabdoviridae*, *Flaviviridae*, *Bornaviridae* and *Togaviridae* (adapted from ViralZone [7]). WNV: West Nile virus; TBEV: Tick-Borne encephalitis virus; LIV: Louping ill virus; JEV: Japanese encephalitis virus; EEEV: Eastern equine encephalitis virus; VEEV: Venezuelan equine encephalitis virus; WEEV: Western equine encephalitis virus, DS: double-stranded, SS: single-stranded.

**Figure 2 viruses-12-00023-f002:**
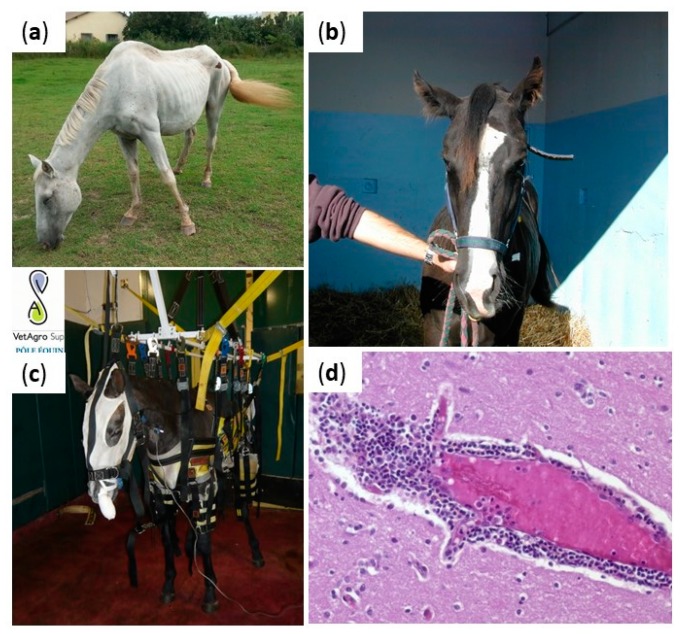
Clinical manifestations and lesions in viral equine encephalitis. Horses infected with equine encephalitis viruses may experience posture deficits (increasing of the lift polygon in (**a**)), cranial nerve deficits (facial paralysis in (**b**)), balance deficiencies (slings in (**c**) can be used to support paretic horses and avoid long and poor prognosis recumbency). Brain lesions are non-specific and include perivascular infiltration of inflammatory cells, observed in (**d**). Credits: Pr Agnès Leblond, VetAgroSup, and Dr Eve Laloy, French Veterinary School of Alfort.

**Figure 3 viruses-12-00023-f003:**
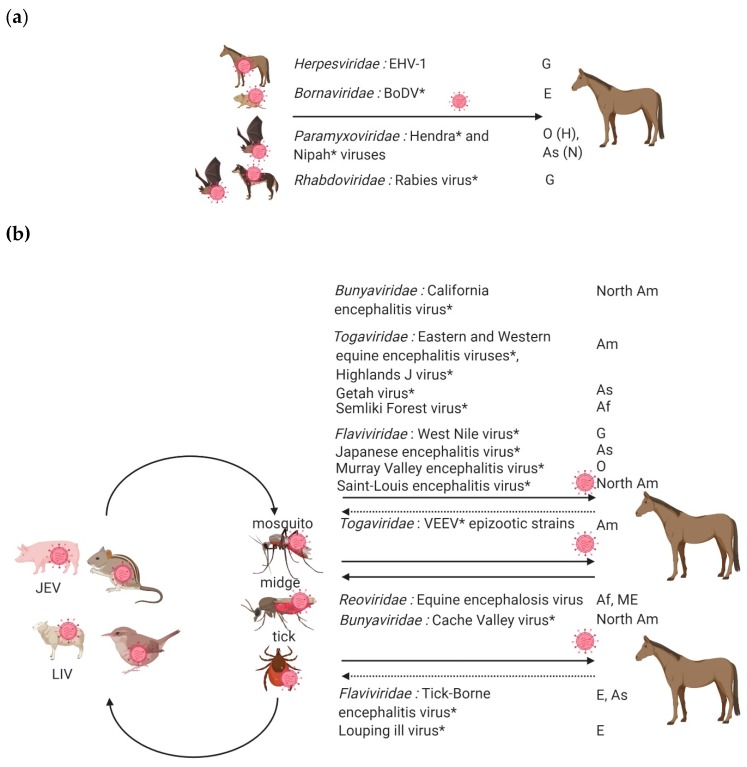
Encephalitis viruses in equines. Transmission mode (direct transmission in (**a**) or arthropod-borne transmission in (**b**)), zoonotic potential (zoonotic viruses are marked with an asterisk) and geographical distribution (Af for Africa, Am for America, As for Asia, E for Europe, ME for Middle East, O for Oceania and G for global) are presented. EHV-1: Equid alphaherpesvirus 1; BoDV: Borna disease virus; VEEV: Venezuelan equine encephalitis virus. Black arrows represent established virus transmission between the two partners. Doted arrows indicate limited virus transmission possibility from the infected horse to its reservoir, with the exception of midge-borne arboviruses and of the mosquito borne VEEV epizootic variants for which horses serve as amplifier hosts. The figure was prepared on BioRender [19].

**Figure 4 viruses-12-00023-f004:**
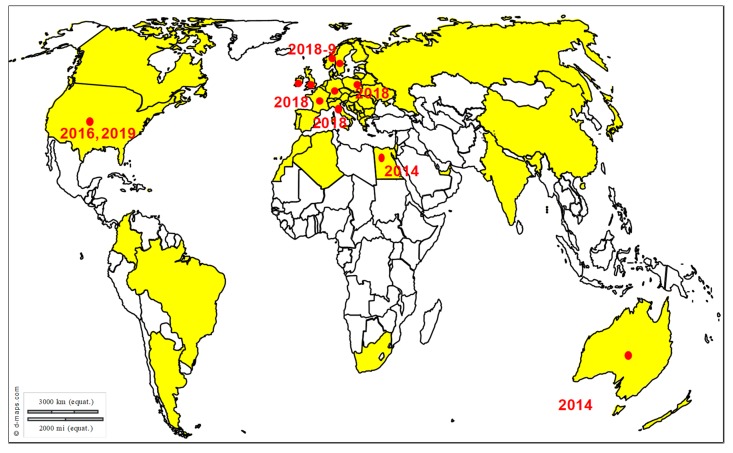
Distribution of EHV-1 outbreaks. Recent outbreaks in horses reported to the OIE WAHIS (World Animal Health Information System) interface [15], Promed Alerts (http://www.promedmail.org) [16] or in the scientific literature are depicted by dots (2014–2019).

**Figure 5 viruses-12-00023-f005:**
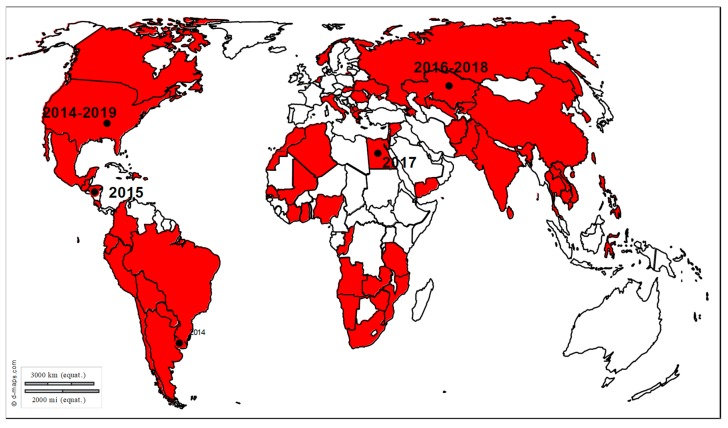
Distribution of rabies outbreaks. Recent outbreaks in horses reported to the OIE WAHIS interface [15], Promed Alerts [16] or in the scientific literature are depicted by dots (2014–2019).

**Figure 6 viruses-12-00023-f006:**
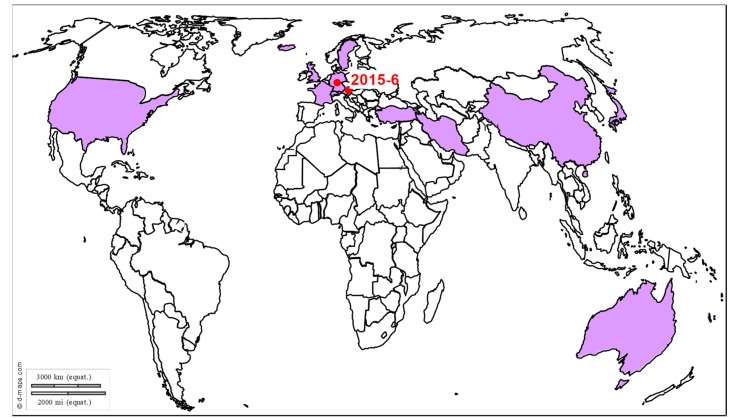
Distribution of Borna disease outbreaks. Recent outbreaks in horses reported to Promed Alerts [16] or in the scientific literature are depicted by dots (2014–2019).

**Figure 7 viruses-12-00023-f007:**
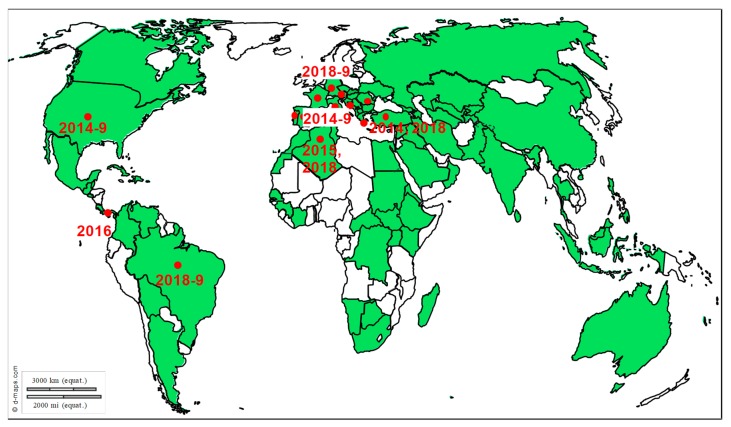
Distribution of WNV outbreaks. Recent outbreaks in horses reported to the OIE WAHIS interface [15], Promed Alerts [16] or in the scientific literature are depicted by dots (2014–2019).

**Figure 8 viruses-12-00023-f008:**
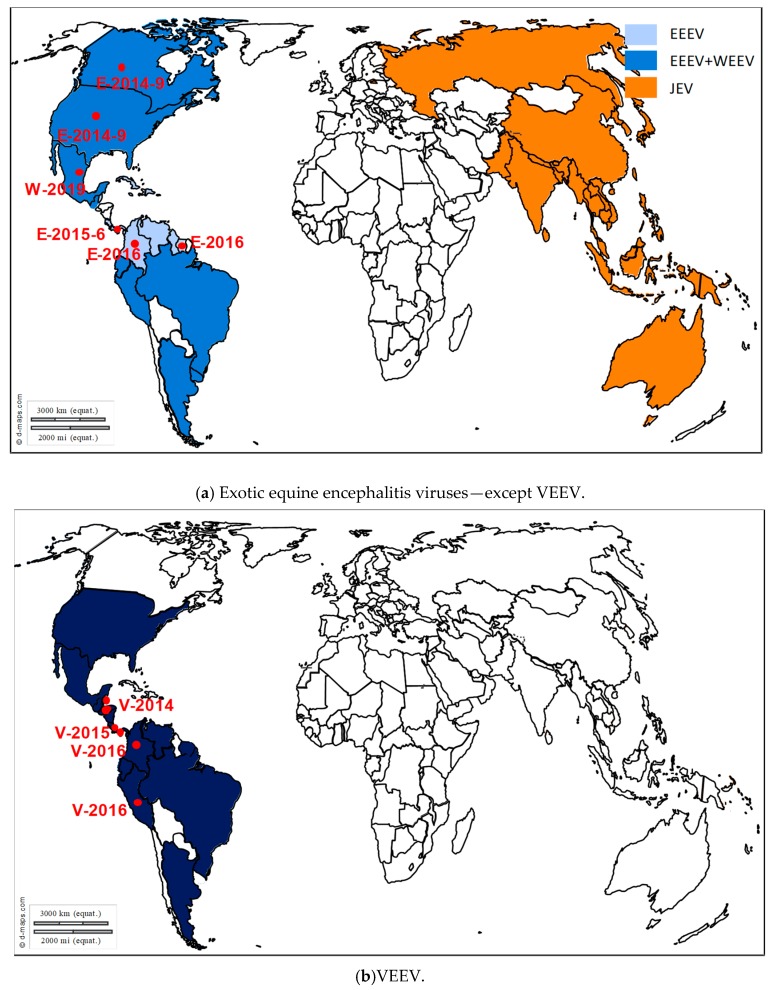
Distribution of exotic equine encephalitis outbreaks: Japanese encephalitis, western and eastern equine encephalitis in (**a**), venezuelan equine encephalitis in (**b**). Recent outbreaks in horses reported to the OIE WAHIS interface [15], Promed Alerts [16] or in the scientific literature are depicted by dots (2014–2019).

**Table 1 viruses-12-00023-t001:** List of diagnostic assays available against equine neuropathogenic viruses enzootic in Europe. PCR = polymerase chain reaction; RT = reverse transcription; VNT = virus neutralization tests; CFT = complement fixation test; DFA = direct fluorescence assay; dRIT = direct rapid immunohistochemistry test; IHC = immunohistochemistry; IFA = indirect fluorescence assay; HIA = hemagglutination inhibition assay; MIA = multiplex immunoassay.

Virus	Diagnostic Assays	Advantages and Shortcomings
EHV-1	Direct assays: PCR, virus isolation	Direct virus detection and typing (SNP-PCR) is possible from easily accessible samples (nasal swabs and blood).
	Serology: VNT, CFT or ELISA	Due to highly prevalent and lifelong infection, diagnostic assays should be interpreted with care. Serology will be informative if serial serum samples can be obtained.
Rabies virus	Direct assays: DFA, dRIT, RT-PCR	Direct virus detection is possible only from the brain of dead animals.
BoDV	Direct assays: RT-PCR, IHC	Due to limited antibody response induced after BoDV infection, definitive diagnostic will be made only after direct virus detection from the brain of dead animals.
WNV/Flaviviruses	Indirect assays preferred: ELISA, IFA, HIA, VNT	Rapid serological screening tests (competition ELISA, IFA) are very sensitive but present a low diagnostic specificity; they should be interpreted with care and confronted with results from confirmatory serological assays (VNT, MIA).
	Direct assays: RT-PCR, virus isolation	Direct virus detection is possible from the brain of dead animals and when positive, indicates recent virus infection.

**Table 2 viruses-12-00023-t002:** List of vaccines licensed in Europe against equine neuropathogenic viruses. Vaccine types and recommended vaccination protocols are presented.

Virus	Vaccine Types Available in Europe	Protection Provided
EHV-1	Inactivated: BIOEQUIN^®^ H (BIOVETA), PNEUMEQUINE^®^ (Boehringer Ingelheim), EQUIP^®^ EHV 1,4 (Zoetis) Live attenuated: PREVACCINOL^®^ (MSD Animal Health), licensed in Germany	Insufficient individual protection against EHM but allows for decreased virus transmission in the vaccinated population, after 2 primes at a 1-month interval (3–4 months with the live attenuated vaccine) and boosts every 6 to 12 months.
Rabies virus	Inactivated: ENDURACELL^®^ R MONO and VERSIGUARD^®^ Rabies (Zoetis), NOBIVAC^®^ Rabies (MSD), RABIGEN^®^ mono (Virbac) and RABISIN^®^ (Boehringer Ingelheim)	Good protection at the individual level provided after a unique prime and boosts performed every year or every 2 years.
WNV	Inactivated: EQUIP^®^ WNV (Zoetis) Recombinant: EQUILIS^®^ West Nile (MSD) and PROTEQ^®^ West Nile (Boehringer Ingelheim)	Good protection at the individual level provided after 2 primes at a 1-month interval and boosts performed every year.

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
