# Peer review of "Viral Equine Encephalitis, a Growing Threat to the Horse Population in Europe?"

_viruses, 2019, doi:10.3390/v12010023_

Round 1
Reviewer 1 Report
I enjoyed reading the review and found it to be a good summary of neurotropic infections of equine. However, the review fell short in not providing tabled summaries on (1) current diagnostic tests with comments on short comings and (2) current vaccines. the information is written and it should not take much for the authors to remedy this gap. Secondly, the authors need to make a careful review of typos (for example Ukraine is misspelled) and edit the manuscript for wordiness.
Author Response
We are grateful to the reviewers for their helpful comments. We carefully considered the comments of the four reviewers and brought the corrections to the manuscript as suggested (in revision mode in the .docx document):
Reviewer 1
I enjoyed reading the review and found it to be a good summary of neurotropic infections of equine. However, the review fell short in not providing tabled summaries on (1) current diagnostic tests with comments on short comings and (2) current vaccines. the information is written and it should not take much for the authors to remedy this gap. Secondly, the authors need to make a careful review of typos (for example Ukraine is misspelled) and edit the manuscript for wordiness.
Two tables, on current diagnostic tests (line 149) and on current vaccines (line 155) in use in Europe have been added, as suggested.
Ukraine has been corrected (line 357), as well as other typos identified along the text (lines 28, 158, etc).
Reviewer 2 Report
The manuscript Viruses-619002 “Viral equine encephalitis, a growing threat for the horse population in Europe?” presents a review of the major virological, clinical and epidemiological features of the main neurotropic viruses inducing encephalitis in equids in Europe and the authors discuss whether these viruses could pose a threat to equidae in this continent.
General comments:
The manuscript is well written overall, but the virological and clinical aspects of the main neurotropic viruses inducing encephalitis in equids has been well and carefully described by other authors in scientific papers and book chapters in the last years. The only new aspect in the present paper is some information on epidemiological of these virus infections in Europe (e.g. WNV in page 12).
In view of the modest addition of new knowledge to the field of viral equine encephalitis I would not recommend publication of this manuscript in its current format in Viruses.
Minor comments:
Lines 28-29 the word “industry” appears double.
Line 30: avoid personal pronouns “we”
Line 31 in the Abstract and other parts of the manuscript “herpes virus” should be replaced by “herpesvirus”.
In Figure 1 (B) of page 3 the virus families are repetitive and confusing. Group the different viruses within each family as done in Figure 3. Also, review Figure 1 for acronyms and italicize virus names.
Line 88: Please spell-out 'CSF' at first use.
Line 140: “EHV-1” appears double.
Lines-203-204: the phrase "with all the clinical information available by the practitioner" appears twice.
Line 217-218- sentence does not make sense: "There is an urgent need to screen and separate sick horses from exposed and potentially infected ones and non-exposed and healthy animals".
Lines 479-480: "TBEV is the most important human tick-borne pathogen in Europe and Asia. The estimated annual incidence rate is 10,000 human infections with a case-fatality rate ranging from 1 to 20%" What is the incidence/prevalence for equids? Detail here TBEV for horses.
References are missing for some sentences (e.g. lines 51-53; lines 124-134; lines 436-437).
Review the location of figures within the text.
Figure 4 needs better resolution.
The authors do not mention about vector diversity/density and their influence on the transmission of equine encephalitis in Europe. Wouldn't that be relevant?
Author Response
We are grateful to the reviewers for their helpful comments. We carefully considered the comments of the four reviewers and brought the corrections to the manuscript as suggested (in revision mode in the .docx document):
Reviewer 2
The manuscript Viruses-619002 “Viral equine encephalitis, a growing threat for the horse population in Europe?” presents a review of the major virological, clinical and epidemiological features of the main neurotropic viruses inducing encephalitis in equids in Europe and the authors discuss whether these viruses could pose a threat to equidae in this continent.
General comments:
The manuscript is well written overall, but the virological and clinical aspects of the main neurotropic viruses inducing encephalitis in equids has been well and carefully described by other authors in scientific papers and book chapters in the last years. The only new aspect in the present paper is some information on epidemiological of these virus infections in Europe (e.g. WNV in page 12).
In view of the modest addition of new knowledge to the field of viral equine encephalitis I would not recommend publication of this manuscript in its current format in Viruses.
We feel that our review covers the main viruses responsible for viral equine encephalitis, while previous reviews mainly considered one or a limited group of such viruses (arboviruses mainly). Updated information on their detection, prevention and transmission risk inside the EU is given.
Minor comments:
Lines 28-29 the word “industry” appears double. Modified
Line 30: avoid personal pronouns “we”. The passive form was used.
Line 31 in the Abstract and other parts of the manuscript “herpes virus” should be replaced by “herpesvirus”. Corrected in lines 32, 118, 170.
In Figure 1 (B) of page 3 the virus families are repetitive and confusing. Group the different viruses within each family as done in Figure 3. Also, review Figure 1 for acronyms and italicize virus names. Figure 1 was modified, as suggested by the reviewer, except from the virus names which were not italicized, according to ICTV prescriptions.
Line 88: Please spell-out 'CSF' at first use. Corrected on line 88
Line 140: “EHV-1” appears double. Modified
Lines-203-204: the phrase "with all the clinical information available by the practitioner" appears twice. Corrected
Line 217-218- sentence does not make sense: "There is an urgent need to screen and separate sick horses from exposed and potentially infected ones and non-exposed and healthy animals". Modified as follows “There is an urgent need to screen and separate potential virus shedders (either confirmed to be infected with EHV-1 or exposed) from non-exposed and healthy animals.”
Lines 479-480: "TBEV is the most important human tick-borne pathogen in Europe and Asia. The estimated annual incidence rate is 10,000 human infections with a case-fatality rate ranging from 1 to 20%" What is the incidence/prevalence for equids? Detail here TBEV for horses. TBEV and LIV infection in horses are very uncommon, as stated lines 581-582. The case fatality rate in equids is not known and only a few seroprevalence studies have been performed. The following data were added in the manuscript: “A few serological surveys in equids are available in the scientific literature. TBEV seroprevalence rates of 20-30% among asymptomatic horses have been reported in Austria and Germany while lower ones have been reported in the Balkans (3-5% in Serbia and Slovakia).”
References are missing for some sentences (e.g. lines 51-53; lines 124-134; lines 436-437). The following references were added: Kumar et al, Open Virol J, 2018 at lines 51-53; Chapman et al, Equine Vet J, 2018 and Weaver et al, Antiviral Res, 2010 at lines 124-134; Beck et al, IJERPH, 2013 at lines 436-437.
Review the location of figures within the text. The different figures have been moved to different sections of the document, to take into account recommendations from reviewers 2 and 4.
Figure 4 needs better resolution. Figure 4 is now provided at a higher resolution and has been splitted in 5 figures in accordance with reviewer 4 comments.
The authors do not mention about vector diversity/density and their influence on the transmission of equine encephalitis in Europe. Wouldn't that be relevant? These aspects are difficult to review, since many information are lacking or partial. To discuss these aspects, we included the followings in the conclusion “A high diversity of mosquito species have been reported in Europe (mainly from Aedes, Culex and Culiseta genera), and highly invasive Aedes albopictus and Ae. japonicus have rapidly established in several European countries over the last decade. Vector competence of native and invasive European mosquito species for equine encephalitis viruses, other than WNV and JEV, has been unfrequently evaluated. Consequently, identification of European regions at risk for the spread of exotic equine encephalitis viruses is difficult and mainly relies on information on mosquito and animal hosts density and on records of opportunistic mosquito species.”
Reviewer 3 Report
the review "Viral equine encephalitis, a growing threat for the horse population in Europe?" describes different neurologic viral diseases in horses. In the abstract and introduction, the authors stated that they will focus on viral disease causing neurological disorders in horses in Europe. However, while going through the review, it was not clear "confusing in different places" if the review "really" on these diseases in horses in Europe.
the authors put a lot of information, which is a great effort. However, with fine organization and focus, it would be better.
comments:
1- the authors stated in different places "neurotropic" viruses. Not all the viruses, mentioned here are neurotropic. For example, EHV-1 is not neurotropic. it is endotheliotropic and neuropathogenic. the virus is not replicating in nervous tissue as the author stated.
2- I would encourage the authors to restructure the review to clearly talk about the same points in each virus in the same order; the authors might use sub-titles if needed. for example: short intro about the virus, virus description (structure), epidemiology and transmission, pathogenesis and clinical signs, diagnosis, control.
3- mainly with enzootic flaviviruses and exotic viruses, but elsewhere: the authors put many details in virus infection in humans and other mammals and barely talk about horses. it would be interesting to read about these diseases in horses in Europe.
4- Hendra and Nipah are not recorded in Europe but mentioned here.
Author Response
We are grateful to the reviewers for their helpful comments. We carefully considered the comments of the four reviewers and brought the corrections to the manuscript as suggested (in revision mode in the .docx document):
Reviewer 3
The review "Viral equine encephalitis, a growing threat for the horse population in Europe?" describes different neurologic viral diseases in horses. In the abstract and introduction, the authors stated that they will focus on viral disease causing neurological disorders in horses in Europe. However, while going through the review, it was not clear "confusing in different places" if the review "really" on these diseases in horses in Europe.
The authors put a lot of information, which is a great effort. However, with fine organization and focus, it would be better.
comments:
1- the authors stated in different places "neurotropic" viruses. Not all the viruses, mentioned here are neurotropic. For example, EHV-1 is not neurotropic. it is endotheliotropic and neuropathogenic. the virus is not replicating in nervous tissue as the author stated. This term ‘neurotropic’ was indeed erroneously used for EHV-1, as well as for Equine Encephalosis Virus. Virus pathogenesis has been described in the review and neurotropic was replaced by neuropathogenic when needed in the abstract (line 30), introduction (line 54, 61, 84, 133), as well as in EHV-1 and EEV sections (lines 186 and 791).
2- I would encourage the authors to restructure the review to clearly talk about the same points in each virus in the same order; the authors might use sub-titles if needed. for example: The same structuration of virus sections was used, as recommended : after a short general introduction on the the disease and virus, virus description, transmission and epidemiology, pathogenesis and clinical signs, diagnosis, prevention and control) are covered.
3- mainly with enzootic flaviviruses and exotic viruses, but elsewhere: the authors put many details in virus infection in humans and other mammals and barely talk about horses. it would be interesting to read about these diseases in horses in Europe. Recent information on equine WNV (lines 541-543), JEV, SLEV and MVEV (lines 690-691), WEEV(lines 725-726) and EEEV (lines 742-746) infections in horses have been added. Recent equine outbreaks (2014-2019) are also depicted in figures 7-8 (ex figure 4) for most enzootic flaviviruses and exotic viruses.
4- Hendra and Nipah are not recorded in Europe but mentioned here. The corresponding paragraph was deleted. Even though Hendra and Nipah are emerging viruses, their transmission through direct or indirect contacts with fruit bats (that have been identified in Australasia and Africa only) make it unlikely that they spread into Europe.
Reviewer 4 Report
In this paper the Authors review the main viral encephalitis of horses. The article includes many bibliographic references and describes the state of the art of the most important viral equine encephalitis.
The topic is interesting and this is an interesting scholarly article. However, I think that some parts should be improved.
In particular, the first part of the article seems a little bit confusing. While sections 2.4 and 3 are well organized and structured, the previous sections do not follow a fixed scheme of presentation, tas commonly used when infectious diseases are described (e.g. aetiology, epidemiology, clinical presentation, diagnosis, prevention). For example, in section 2.2 some epidemiological information are reported (lines 340-348) between diagnosis (lines 330-339) and prevention (349-353). Similarly, in section 2.3 epidemiology is described almost at the end of the section instead of in the first part of the section, as commonly done. I suggest to rearrange sections 2.1, 2.2 and 2.3 similarly as section 2.4.
In addition, the Authors should evaluate if it is possible to move figures reporting data of diseases described in different sections in a common preliminary section instead of in the specific section 2.1. Equid Herpesviruses.
Please, refer to the most recent virus taxonomy (2018).
Although reference section includes an excellent selection of articles, some important treference are missing (listed in the specific comments).
Specific comments:
29: please check if “industry” is erroneously repeated
31: herpesvirus instead of herpes virus
34: West Nile virus instead of west nile virus
61: Fig 1 is a complex picture reporting transmission mode, zoonotic potential and geographical distribution. On the contrary, the reference Figure 1 is reported at the end of sentences that are not describing transmission mode, zoonotic potential or geographical distribution. In my opinion this picture should be not inserted at this point but at the end of a paragraph about transmission, distribution and zoonotic potential. Figure 1 describes contents that are not reported in the text before line 61.
61-64: I suggest to move these lines at the end of this section and immediately before the next section, togheter with lines 121-122
132-133: please check the spelling of the diseases
142: please, refer to the most recent virus taxonomy (https://talk.ictvonline.org/taxonomy/ ). For example, the correct name of Equid Herpesvirus 8 is Equid alphaherpesvirus 8. The same is for the other herpesviruses. Expecially in reviews, which often are scholarly articles, the use of the current taxonomy is really important.
146: please check the taxonomy and the name of the virus
147: abbreviation of EHM has been already reported at line 87
156: the name of the subfamily should be in italics
158: I do not agree with this sentence. Latent infection is not established only in peripheral blood leukocytes. Please, check papers describing other sites of latency (e.g. Slater JD, Borchers K, Thackray AM, Field HJ. The trigeminal ganglion is a location for equine herpesvirus 1 latency and reactivation in the horse. J Gen Virol. 1994 Aug;75 ( Pt 8):2007-16.; Baxi MK, Efstathiou S, Lawrence G, Whalley JM, Slater JD, Field HJ. The detection of latency-associated transcripts of equine herpesvirus 1 in ganglionic neurons. J Gen Virol. 1995 Dec;76 ( Pt 12):3113-8.
207: a point is required
263: although figure 4 is very interesting, I think that it is not correct to insert data regarding other diseases in the section describing EHV disease. It would be correct if in this section the distribution of all diseases was described. In the present case, only data on EHV disease should be reported here.
289: the name of the viruses should be in italics
289-295: the list of viruses is not complete and the most recent references about this topic have been not considered. For example, Gannoruwa bat lyssavirus is not reported in the list. Please, check the references: Panduka S. Gunawardena1, Denise A. Marston1, Richard J. Ellis, Emma L. Wise, Anjana C. Karawita, Andrew C. Breed, Lorraine M. McElhinney, Nicholas Johnson, Ashley Banyard, and Anthony R. Fooks. (2016) Lyssavirus in Indian Flying Foxes, Sri Lanka. Emerging infectious diseases. 22(8), 1456-1459; Hayman DT, Fooks AR, Marston DA2, Garcia-R JC. The Global Phylogeography of Lyssaviruses - Challenging the 'Out of Africa' Hypothesis. PLoS Negl Trop Dis. 2016 Dec 30;10(12):e0005266. doi: 10.1371/journal.pntd.0005266. eCollection 2016 Dec.
345-346: a space is required after comma
357: please refer to the most recent virus taxonomy made in 2018 (https://talk.ictvonline.org//taxonomy/p/taxonomy-history?taxnode_id=201851553)
422: preventive measures or availability of vacines are not discussed in this section
426: the correct name of the virus is Louping ill virus
451-457: reference is required
467-468: reference is required
582-591: references are required
603: reference 145 describes the finding of JEV sequences in a pool of mosquito (Culex pipiens). The reference about the finding of JEV sequences in birds in Italy is: Preziuso S, Mari S, Mariotti F, Rossi G. Detection of Japanese Encephalitis Virus in bone marrow of healthy young wild birds collected in 1997-2000 in Central Italy. Zoonoses Public Health. 2018 Nov;65(7):798-804. doi: 10.1111/zph.12501. Epub 2018 Jul 4.
616: do you mean EEE, WEE and VEE instead of EEEV, WEEV and VEEV?
618: do you mean WEE?
639: please evaluate to change will be with “are” or “can be”
Round 2
Reviewer 2 Report
After review made by the authors the article should be considered for publication in Viruses.
General Comments:
For information on how to write virus and species names please refer to ICTV at: https://talk.ictvonline.org/information/w/faq/386/how-to-write-virus-and-species-names
Author Response
We carefully reviewed the nomenclature used for virus species and names in our review and corrected a few mistakes that had not been corrected earlier.
Reviewer 3 Report
the review was significantly improved. Almost all the comments are addressed and accepted.
I have minor comments:
1- some of the figures are duplicated such as Fig. 3 and "i guess the deleted old Fig. 4"
2- Table 2: for the table to be pretty: it is better to just mention the name of the test without comments. you can add another column to describe the test. Maybe divide the last column into two (advantages and shortcomings).
3- the authors used "etc" at different places in the text. I encourage the author to be precise and avoid using this word as it is not informative.
4- page 8 line 195: it is "Alphaherpesvirinae"
Author Response
-The duplicate pictures had been deleted in revision mode in the Word document and were still visible. We deleted them completely in the version submitted today.
-Table 1 was modified as proposed, and comments were deleted from the colums "Diagnostis assays" and transfered in the third column. No other column was added in the table.
-"etc' was withdrawn from the text : corresponding sentences were modified or completed lines 132, 133, 160, 468, 589, 603, 758
-"Alphaherpervirinae" corrected in the text line 195
Reviewer 4 Report
Most comments I suggested in the previous revision round have been addressed. Viruses nomenclature should still revised because mistakes are still present (e.g. Japanese encephalitis virus instead of Japanese Encephalitis at line 59, etc.).
I don't feel qualified to judge about the English language and style.
Although the scientific contents of the article are moderately relevant, in my opinion this open access article can be interesting as scholarly article.
After checking the viruses nomenclature, the article can be accepted for publication in Viruses.
Author Response
We carefully reviewed the nomenclature of virus species and names according to https://talk.ictvonline.org/information/w/faq/386/how-to-write-virus-and-species-names and corrected mistakes that were not seen earlier; more specifically at lines 58-61, 71-73, 125, 195, 479, 653-6.